# A Narrative Review: Syndecans in Aortic Aneurysm Pathogenesis and Course—Biomarkers and Targets?

**DOI:** 10.3390/ijms26031211

**Published:** 2025-01-30

**Authors:** Calogera Pisano, Laura Asta, Adriana Sbrigata, Carmela Rita Balistreri

**Affiliations:** 1Cardiac Surgery Unit, Department of Precision Medicine in Medical Surgical and Critical Area (Me.Pre.C.C.), University of Palermo, 90134 Palermo, Italy; adriana.sbrigata@gmail.com; 2Department of Cardiac Surgery, Clinical Mediterranean, 80122 Naples, Italy; astalaura92@gmail.com; 3Cellular, Molecular and Clinical Pathological Laboratory, Department of Biomedicine, Neuroscience and Advanced Diagnostics (Bi.N.D.), University of Palermo, 90134 Palermo, Italy

**Keywords:** endothelial glycocalyx, syndecans, aortic aneurysm

## Abstract

The maintenance of the integrity of the entire endothelium, glycocalyx included, and, therefore, of tissue aorta’s homeostasis, depends on the expressions of several molecular pathways and their interactions, such as syndecan molecules. Alterations in syndecans, i.e., quantitative alterations or linking to their shedding, contributes to invoking endothelium dysfunction, which causes damage to the vessel wall due to the increased production of growth-stimulating and pro-inflammatory gene products. Inflammatory processes negatively affect the integrity of the endothelial glycocalyx, a dynamic layer of the luminal portion of endothelial cells composed of proteoglycans, glycoproteins, and glycosaminoglycans, i.e., syndecans. In turn, structural alterations in the endothelial glycocalyx influence the coagulative state, increasing pro-thrombotic processes. The family of syndecans constitutes a major component of glycocalyx or, more accurately, the major source of cell surface heparan sulfate. It encompasses four components: syndecan-1, syndecan-2, and syndecan-4 (with syndecan-3 only expressed in neural tissue), which have a fundamental role in regulating the events of acute and chronic aorta damage subsequently correlated with the formation of aneurysms. As such, the aim of our review is to highlight the current knowledge on the roles of syndecans and to analyze their relationship with the pathological processes of the aortic wall based on the most recent literature.

## 1. Introduction

Aortic aneurysm is a usually silent clinical condition that progresses slowly until it reaches a critical point, beyond which the progression of an increasing aortic diameter can be very rapid, leading to dissection or rupture [1,2]. For this reason, the 5-year survival of untreated patients is 10–20% due to the high rate of lethal ruptures [3].

An aneurysm is defined as a permanent, localized arterial dilation greater than 1.5 times the normal diameter of the vessel that involves all three layers of the aortic wall (different from pseudoaneurysm).

The dimensions of the thoracic aorta vary according to age, sex, and body surface area (BSA): they increase progressively with advancing age and weight gain, while at the same age, men have a larger diameter than women [4].

Therefore, there are various methods used in both the clinical and scientific fields, which allow the dimensions of the aortic diameter to be related to the individual, such as the Z-score (used mainly in the pediatric field) and the aortic size index, which allows the risk of dissection to be stratified in adults based on the aortic diameter in relation to the height and weight (BSA) of the subject [5] (Figure 1).

Unlike abdominal aortic aneurysm, which demonstrates etiopathogenesis mostly related to atherosclerotic pathology, in thoracic aortic aneurysm, the etiology is extremely variable.

Therefore, based on the responsible etiological agent, we can distinguish different categories (sporadic, syndromic, and familial non-syndromic aneurysms).

In the case of sporadic form, the precise pathogenetic processes related to the degeneration of the medial tunica and remodeling of the entire aorta wall are not fully known [3,6].

Microscopically, the aortic wall can be divided into three layers: intima, media, and adventitia (Figure 2).

The intima is the most superficial layer of the aortic wall, the one that faces the vascular lumen. It is composed of a single layer of endothelial cells resting on a basement membrane and supported by an elastic lamina that marks the boundary with the media. The internal elastic lamina is composed of type IV collagen and laminin. The primary role of the intima is to provide a smooth, non-thrombogenic surface for blood flow, and it also regulates vascular tone (through the secretion of prostacyclin and endothelium-relaxing factors) and participates in inflammatory and immune responses.

The media is the central layer of the aortic wall and is composed of collagen fibers, elastic fibers, smooth muscle cells, and other molecules organized into concentric lamellar units. Concentric sheets of elastin enclose smooth muscle cells and are connected by finer elastic fibers. Collagen fibers are also spaced and aligned circumferentially. In the normal aortic root, the walls are made up of 50–70 layers of concentric lamellar units. This elastic and muscular organization allows the aorta to expand and counteract the pulsatile blood flow generated by the heart (Figure 3).

The adventitia is the outermost layer of the aortic wall and consists of multiple layers of collagen fibers, especially type I collagen, mast cells, and fibroblasts that provide additional support to the vascular structure.

However, in the past few years, our group demonstrated the role of diverse pathways, differentially expressed, including, for example, Notch, TLR4, eNOs, RAS, TGF-β, and MMP pathways, as well as the crucial involvement of inflammatory pathways [8,9,10,11]. They were demonstrated to induce the remodeling of the aorta wall, including the infiltration of leukocytes, increased levels of systemic parameters related to chronic inflammation, and the augmented release of enzymes able to degrade the aortic wall, and to determine thoracic aortic aneurysm formation [12,13,14].

Recently, numerous data in the literature demonstrated how the alteration in the endothelial glycocalyx (eGCX) increases inflammatory processes and leukocyte adhesion, which augment the risk of vascular diseases [15,16,17,18,19]. The endothelial glycocalyx is a proteoglycan complex, composed of a protein core (syndecans) and glycosaminoglycans, such as heparan sulfate, chondroitin sulfate, and hyaluronas, lining the luminal surface of endothelial cells (ECs) in all blood vessels [20,21] (Figure 4).

The integrity of the endothelial glycocalyx is essential for maintaining the homeostasis of the cardiovascular system, as it plays an important role in several functions of blood vessels, semi-permeable barrier, shear stress mechanosensation, and mechano-transduction, stimulating the production of nitric oxide, protecting from cellular infiltration and the activation of inflammatory processes [23].

Thus, the aim of our review is to analyze the role of endothelial glycocalyx dysfunction, and the relationship of syndecans, in aortic aneurysm (AA) pathogenesis. This might help to identify new biomarkers and targets.

## 2. Alterations in the Endothelial Glycocalyx Underlying Various Pathological Processes

It was previously adequately demonstrated that endothelial cell alterations play a key role in the pathogenesis of various microcirculatory alterations that then lead to the onset of systemic pathologies. In particular, major alterations affect the metabolism of nitric oxide (NO), with a consequently reduced bioavailability of the same and altered vasodilation [24].

Furthermore, the flowmetric alterations determine, through a mechano-transduction mechanism, an increase in inflammatory processes, pre-repression of atherosclerotic phenomena, and an increase in thrombogenic processes [25].

There are numerous molecular targets of the mechanotransduction processes that lead to the aforementioned pathological alterations. Among these, an important role is played by the endothelial glycocalyx.

As already mentioned, the endothelial glycocalyx is a proteoglycan complex, composed of a protein core (syndecans) and glycosaminoglycans, such as heparan sulfate, chondroitin sulfate, and hyaluronas, lining the luminal surface of endothelial cells (ECs) in all blood vessels; and it seems that the luminal portion of the endothelial glycocalyx is the one involved in the mechanotransduction processes resulting from fluxmetric alterations.

In particular, damage to this complex, especially by pro-inflammatory cytokines that increase its degradation, leads not only to an alteration in tissue permeability, since the endothelial glycocalyx has an essential role in vascular permeability, allowing the passage of water and solutes and acting as a passive barrier to the passage of proteins, as well as to reduced protection from shear stress, which itself is a regulator. One of the most significant effects of the alteration in the endothelial glycocalyx is represented by the increased expression of the activity of the endothelial nitric oxide synthase (eNOS), with a consequently reduced bioavailability of NO and the increased formation of peroxynitrite. The endothelial glycocalyx represents, in fact, a barrier to the passage of liquids and proteins mediated by the different effects of hydrostatic pressure and oncotic pressure, in addition to being a sensor of shear stress and pressure, avoiding the hyperactivation of the cell surface receptors. Thus, it was demonstrated that an alteration in the endothelial glycocalyx determines an increase in the cytosolic calcium concentration; the activation of eNOS; Rho family GTPases through the activation of the transcription factors Kruppel-like-2 and nuclear factor-like-2 and tyrosine kinases, Ras, Erk, and JNK; and the phosphorylation of the PECAM-1 intracellular domain, with a consequent increase in tissue vasodilation and failure to adapt blood flow to tissue demands [26].

However, progress in the study of the endothelial glycocalyx has allowed us to further expand our knowledge of the pathogenic processes resulting from its degradation.

It has been seen how it correlates to an increase in oxidative stress (related to the reduced synthesis of NO and increased levels of reactive oxygen species (ROS)), an increase in angiogenic processes (since the endothelial glycocalyx regulates the binding and bioavailability of the main stimulant of angiogenesis VEGF to its receptors), an increased dysregulation of smooth muscle cells (due to the effect of the endothelial glycocalyx on the pathways that regulate proliferation, cell motility, and phenotype change), and an increase in the destruction of the extracellular matrix (due to the increased production of metelloproteinases (MMPs)).

Therefore, since aneurysmal pathology shares the same pathogenetic mechanisms (increased oxidative stress and angiogenic processes, dysregulation of smooth muscle cells, and increased destruction of the extracellular matrix), it is easy to understand how damage to the endothelial glycocalyx is strictly correlated to aneurysmal pathology [21] (Figure 5).

## 3. Syndecans and Their Role in Disease: What Is the Role in Aortic Aneurysm?

Emerging evidence demonstrated the contribution of eGCX, with pleiotropic roles, in the onset and progression of cardiovascular diseases (CVD), aneurysm included.

eGCX dysfunction is characterized by its degradation and has been observed to occur in diverse CVD, such as AA [28]; therefore, circulating levels of related eGCX degradation products have been proposed as CVD biomarkers. In addition to their role as diagnostic biomarkers, some eGCX fragments act as pathogenic factors in disease progression, thus representing potential prognostic biomarkers, leading to the development of pharmacological interventions and potential strategies to attenuate eGCX degradation or restore its integrity, maintaining endothelial health into adult life (Figure 6).

Among the eGCX degradation products, the syndecans (SDCs) and heparan sulfate proteoglycans (HSPGs) have attracted the attention of several researchers [29,30].

The components of the glycocalyx are glycoproteins with short acid oligosaccharides and terminal sialic acids (SAs), oligosaccharides and heparan sulfate proteoglycan (HSPGs), such as SDCs, and glycosaminoglycans (GAGs).

The SDCs family include four members, SDC-1, SDC-2, SDC-3, and SDC-4, consisting of a core protein modified by heparan sulfate (HS) chains.

Each SDC has defined expression patterns and functions in their respective target tissues. SDC3 is expressed in neuronal and musculoskeletal cells.

SDC-3 (N-SDC or *neuronal* SDC) is a transmembrane protein 442 amino acid long. Analysis of SDC-3 in experimental models of inflammation and AD showed that patients with AD overexpressed SDC-3, not only in the brain but also in the periphery. Consequently, SDC-3 could serve as a basis for the development of future AD diagnostics. Syndecans, core proteins of the endothelial glycocalyx, are a family composed of four cell surface proteoglycans (namely, syndecan-1, -2, -3, and -4).

Syndecans interact with a wide variety of molecules, including growth factors, cytokines, proteases, adhesion receptors, and extracellular matrix components, mainly via pendant glycosaminoglycans, such as heparine sulfate and chondroitin sulfate, which sequester and regulate the activity of heparin-binding growth factors, pro-inflammatory chemokines, and proteases [31,32].

### 3.1. SDCs and Their Expression in Inflammatory Conditions

SDC expression depends on the release of growth factors and inflammatory molecules. It has been observed that SDC-1 mRNA levels increase in conditions of improved release of growth factors, such as platelet-derived growth factor (PDGF), FGF-2, and TGF-β. On the contrary, the SDC-1 levels appear reduced in the presence of factors that inhibit cell growth (e.g., IFN-γ).

Moreover, SDC-2 mRNA increases in response to TGF-β1 and IL-1β, while it decreases in response to IFN-γ, and remains stable in the presence of other growth factors [33].

SDC-4 synthesis is positively regulated by FGF-2, TGF-β1, and hypoxia-inducible factor-1 pathway and the p38 MAPK pathway. In the case of endothelial damage, an increased release of FGF-2 occurs and invokes increased cell migration and proliferation with the aim of repairing the damaged site.

However, it has been shown that the increased synthesis of SDC-1 downregulates SDC-4 synthesis by suppressing the ERK1/2 and p38 MAPK signaling pathways to ensure tissue homeostasis [34].

### 3.2. SDC and Aorta Aneurysm Formation

In Table 1, we summarize the data on the role of SDCs in the onset of aorta aneurysms.

In particular, Wen and colleagues studied the expression of SDC-1, -2, -4 using 8-week-old male Apo-E-deficient mice (mice subjected to an atherogenic diet), before and after the stimulation of arterial hypertension by the infusion of angiotensin II with the aim of inducing the formation of aortic aneurysms. Initially, only the presence of SDC-4 was detected within the smooth muscle cells of the aortic wall media, while the presence of SDC-1 and -2 was not detected. After a week of treatment, and especially during the process of aortic aneurysm formation, an increased synthesis of SDC-1 emerged, associated with the infiltration of macrophages, mainly at the level of the periadventitial aorta. In particular, in the created aneurysms, an increase in the expression of SDC-2 was evident, while, due to the fragmentation of smooth muscle cells, a heterogeneous distribution of SDC-4 was observed. Considering these results, the authors hypothesized that the increased release of SDC-1, with associated macrophage infiltration, may stimulate the inflammatory process, which, together with the increase in proteolytic activity, raises the degradation of the ECM, constituting the basic pathogenic process of aneurysm formation [35].

Conversely, SDC-2 enhances the expression of TGF-β I and II [36], resulting in the increased synthesis of collagen and elastin fibers, smooth muscle cell proliferation, and inhibition of metalloproteases. Thus, its increased expression during the aneurysm formation process could be explained as a compensatory mechanism to reduce dilation.

Similar results emerged from the work of Zalghout et al. They conducted an in vitro study on the aortic wall of twenty-five patients undergoing thoracic aortic replacement surgery and compared it with the aortic wall of eleven healthy subjects and emerged via RT-qPCR evaluation, ELISA, and histological analysis. The data that emerged indicated a greater expression of syndecan-1 in the media of patients affected by aneurysms, especially within the smooth muscle cells. Furthermore, they conducted a further in vivo analysis on 3-week-old SDC-1+/+ a SDC 1−/− divided into three groups: the first group without treatment up to 8 weeks of life and the second and third groups treated with β-aminopropionitrile fumarate for 28 days, then infused subcutaneously of angiotensin II. Upon analyzing the onset of aneurysms in the three different groups, they demonstrated an equal incidence of thoracic and abdominal aneurysms in the two different populations; after 3 days from the infusion of angiotensin II, there was no relevant different incidence of aneurysms; after 28 days of treatment, there was an increase in the incidence of thoracic aneurysms in the syndecan-1+/+ population compared to the syndecan-1−/− population, as well as, finally, an increased incidence of abdominal aneurysms (8%) of the SDC 1−/− population compared to the SDC-1+/+ population (no presence of abdominal aneurysm), demonstrating the protective role of SDC-1 in the onset of abdominal aneurysms. Finally, no significant differences were highlighted between the two populations (SDC-1+/+ and SDC-1−/−) in terms of elastin degradation, collagen fiber fragmentation, and leukocyte infiltration (Figure 7) [37].

Likewise, Xiao et al., through the administration of elastin and angiotensin II in SDC 1−/− mice, highlighted during aneurysm formation an increase in macrophages associated with SDC-1, an increase in proteolytic activity, with an increase in MMP-9 and MMP-2 within the aortic wall and, furthermore, the reduction in the expression of IL-10 and Foxp3 demonstrates how the deficit of SDC 1 determines a reduction in the ability to limit the inflammatory process [32].

A key role has emerged for SDC-4 that was demonstrated in the manuscript by Hu and colleagues. In addition to the reduced expression of SDC-4 in the aortic wall of patients with abdominal aneurysms, they emphasized a very high incidence of abdominal aneurysms and a greater degradation of elastic fibers in SDC4−/−Apo E−/− mice treated with angiotensin II infusion, compared to the ApoE−/− group. Furthermore, by constructing SDC4 knockdown (SDC4-KD) and SDC4 overexpression (SDC4-OE) viruses capable of infecting smooth cells, they observed a reduction in the levels of paxillin (cell adhesion protein) in SDC4-KD, an increase in the levels of MMP2 and MMP9, and a reduction in the levels of alpha-SMA, calponin1, and SM-MHC in SDC4-KD treated with angiotensin II in the physiological conditions. On the contrary, no difference was identified in the SDC4-OE population. Similarly, stimulation with angiotensin II determined a rise in the production of IL-1β, IL6, and TNF-α (inflammatory cytokines) both in the control group, but especially in SDC4-KD, while strongly reduced in the SDC4-OE group. Finally, phenotypic changes in VSMCs via the RhoA-F/G-actin-MRTF-A pathway in the SDC4-KD group were found. These lasted by enhancing the secretory characteristics of smooth muscle cells and were further increased the risk of abdominal aneurysms [38].

**Table 1 ijms-26-01211-t001:** The recent data about the relationship of syndecans in aorta aneurysm formation.

Authors, Year	Method	Results
Wen et al., 2007, [35]	8-week-old male Apo-E-deficient mice fed an atherogenic diet, after stimulation of arterial hypertension by infusion of angiotensin II with the aim of inducing the formation of aortic aneurysms	Increased expression of syndecan-1 and -2, inhomogeneous distribution of syndecan-4
Zalghout et al., 2022 [37]	in vitro: RT-qPCR evaluation, ELISA and histological analysis of twenty-five patients undergoing thoracic aortic replacement surgery and compared it with the aortic wall of eleven healthy sub in vivo: 3-week-old SDC-1+/+ a SDC 1−/− mice divided into 3 groups (without treatment; treated with β-aminopropionitrile fumarate for 28 days, then infused subcutaneously of angiotensin II)	Increased expression of syndecan-1 in the media of patients affected by aneurysm; major incidence of thoracic aneurysms in the syndecan-1+/+ population; higher incidence of abdominal aneurysms in SDC 1−/− population; no significant differences in terms of elastin degradation, collagen fiber fragmentation and leukocyte infiltration
Xiao et al., 2012 [32]	in vivo: administration of elastin and angiotensin II in SDC-1−/− mice	increase in macrophages associated with syndecan-1, an increase in proteolytic activity, reduction in the ability to limit the inflammatory process
Hu et al., 2021 [38]	in vitro: immunofluorescence and western blotting detection of syndecan-4 in vivo: SDC4−/−apoe−/− mice treated with angiotensin II infusion compared to the apoe−/− group	reduced expression of syndecan-4 in the aortic wall of patients with abdominal aneurysm; high incidence of abdominal aneurysms and a greater degradation of elastic fibers in SDC4−/−apoe−/− mice treated with angiotensin II; increase in the levels of MMP2, MMP9 and a reduction in the levels of alpha-SMA, calponin1 and SM-MHC in SDC4-KD treated with angiotensin II; increase in the production of IL-1beta, IL6 and TNF-alfa specially in SDC4-KD; phenotypic changes in VSMCs via the RhoA-F/G-actin-MRTF-A pathway in the SDC4-KD group

## 4. Considerations in the Experimental Results Obtained Until Now

Important results were obtained in the above-mentioned studies. However, the limited number of results encourages implementation for clearing all gaps in the research on SDCs in aorta aneurysm and understanding how they can mediate diverse biological effects having differential inflammatory or anti-inflammatory characters, or, in other words, how they can modulate the onset and progression of aorta aneurysm.

The data described evidence of a differential role of SDCs during the medial degeneration and the consequent remodeling of aorta wall, as well as the activation and inhibition of diverse cytokines and pathways. Consequently, the study of epigenetic factors able to influence the levels of expression of SDCs in relation to systemic and microenvironmental conditions, and, for example, in relation to inflammatory conditions, might also be of help. For example, the group of Li and coworkers recently demonstrated the crucial role of miR-17-3p in the interaction of vascular endothelial cells and inflammatory cells, which were, in turn, linked to abodominal aorta aneurysm formation. The prominent miR-17-3p expression has been, indeed, associated with the shear stress, representing a significant induction mechanism significantly linked to the onset of abdonminal aorta aneurysm. However, the precise role of miR-17-3p on and its impact on the glycocalyx remains unclear. Thus, Li and coworkers used astragaloside IV (AS-IV) (a small-molecule saponin (molecular weight = 784) derived from *Astragalus* is noted for its anti-inflammatory properties) in 40 Sprague–Dawley rats with abdominal aorta aneurysm, established using porcine pancreatic elastase. Their aim was to elucidate the AS-IV mechanism of action, focusing on the shedding of the glycocalyx in aortic endothelial cells. The results obtained proved that AS-IV limited aortic damage in such rats by decreasing both the aortic diameter and glycocalyx damage. In addition, AS-IV inhibited the boost in miR-17-3p expression and induced the SDC1 expression. Thus, they affirmed that miR-17-3p may damage the glycocalyx of aortic endothelial cells by targeting SDC-1. AS-IV may raise SDC1 expression by inhibiting miR-17-3p, thereby protecting the glycocalyx and alleviating the onset of aorta aneurysm. In another recent study, Zhang and coworkers demonstrated that SDC-4 is a target of miR-629-5p in the case of pediatric acute respiratory distress syndromes (PARDS). While the group of dos Santos showed that miR-126, in the case of laminar shear stress (LSS) on human endothelial cells (HUVECs), has a role in the up- and downregulation of genes involved in atherosclerosis by invoking high SDC-4 expression. Such evidence demonstrates that further in-depth research is imperative, and it might clear the role as biomarkers and targets of aorta aneurysm.

## 5. Conclusions and Future Perspectives: SDCs as Biomarkers and Targets in Aorta Aneurysms

Therefore, eGCX injury appears to be a fundamental driver in the onset and progression of the diverse CVDs, including AA. This evidence has led researchers to hypothesize that its products of degradation, such as SDCs, as mentioned above, can represent potential biomarkers of CVDs, AA included, despite the limited evidence until now. In addition, the heterogeneous expression of such molecules in the course of AA leads us to propose that SDCs can also constitute optimal prognostic biomarkers. Certainly, further studies are needed to support such clinical applications. Our proposal, which, however, still needs more scientific support, would include the search for a clear laboratory correlation of SDC values with an increased risk of aortic aneurysms through increasingly widespread histopathological research in patients with already known pathology. We believe, in fact, that before considering the alteration in SDC values applicable in the preventive phase of the pathology, more solid scientific evidence is needed to which we hope our manuscript can contribute.

In this regard, a clinical correlation between the levels of certain circulating inflammatory markers and the risk of aortic aneurysms was proposed. In particular, Xu and colleagues conducted a bidirectional Mendelian randomization study, weighted by inverse variance, hepatocyte growth factor (HGF), matrix metalloproteinase-7 (MMP-7), MMP-12, and essential modulator NF-kappa-B (NEMO/IKKγ), and found an association with a potential increased risk of abdominal aortic aneurysms, while the reduced risk for the onset of aortic pathology was correlated with increased levels of platelet-derived growth factor BB (PDGFbb), interleukin-4 (IL-4), IL-12p70, IL-10, IL-6Rα, and myeloperoxidase (MPO), i.e., risk of AAA [39]. Similarly, based on the now recognized crucial role of the inflammatory process in the pathogenesis of abdominal aortic aneurysm, the Norwegian group led by Håland conducted a prospective cohort study (46,322 participants) in which they performed a Cox proportional hazards regression to examine the associations between HS-CRP (high-sensitivity C-reactive protein) and the risk of abdominal aortic aneurysm, demonstrating that individuals with HS-CRP 2 mg/L or higher had almost double the risk of developing aortic disease compared to individuals with HS-CRP less than 2 mg/L [40].

Furthermore, the researchers suppose that they can represent optimal targets to develop new approaches and therapeutic strategies to restore eGCX and its functions. Accordingly, an approach might be represented by the reduction in the activity of cytokines and leukocyte extravasation, considered an emerging therapeutic strategy in limiting tissue-damaging inflammatory responses and restoring immune homeostasis in inflammatory diseases, AA included. For example, it was proven that soluble forms of SDC-1 exert helpful anti-inflammatory effects by removing chemokines, suppressing proinflammatory cytokine expression and leukocyte migration, and inducing the autophagy of proinflammatory macrophages [41]. Furthermore, an association has been shown between an increase in the enzymatic release of syndecan-1 in trauma patients upon admission and increased sympathoadrenal activity with consequently increased mortality rate. This finding, therefore, suggests the use of syndecan-1 values in the clinical management of trauma patients, as increased syndecan-1 levels correlate with a more severe clinical picture [42].

In contrast, endogenous SDC-2 appears to exert proinflammatory effects, and SDC-4 appears to mediate beneficial anti-inflammatory effects and regulate Hh and Wnt signaling pathways involved in systemic inflammatory responses. Taken together, targeting the vascular eGCX-derived products, such as soluble SDC-1, SDC-2, and SDC-4, might represent a potential therapeutic strategy for suppressing overstimulated cytokine and leukocyte responses in the aortic wall by limiting or stopping the consequent degeneration and remodeling.

## Figures and Tables

**Figure 1 ijms-26-01211-f001:**
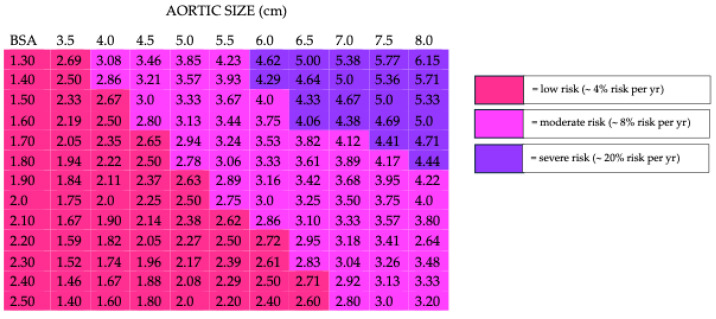
Aortic size index: risk of dissection stratified in adults based on the aortic diameter in relation to the height and weight (BSA) of the subject.

**Figure 2 ijms-26-01211-f002:**
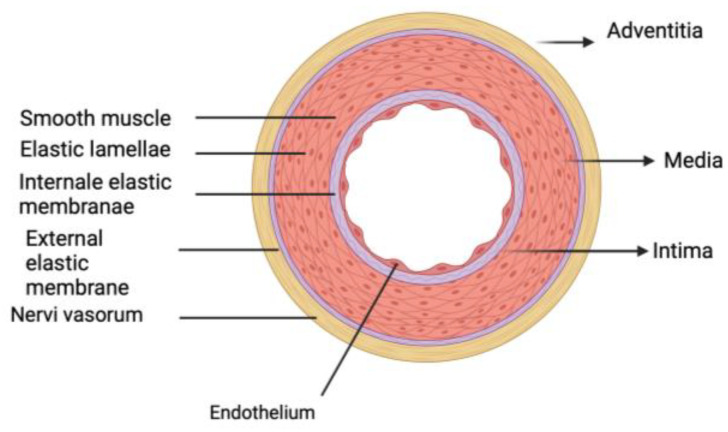
Aortic wall can be divided into 3 layers: intima, media, and adventitia.

**Figure 3 ijms-26-01211-f003:**
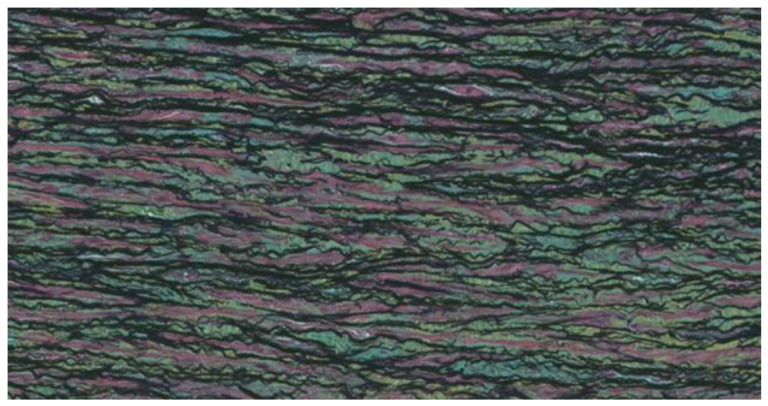
The media is the central layer of the aortic wall and is composed of collagen fibers, elastic fibers, smooth muscle cells, and other molecules organized into concentric lamellar units [7].

**Figure 4 ijms-26-01211-f004:**
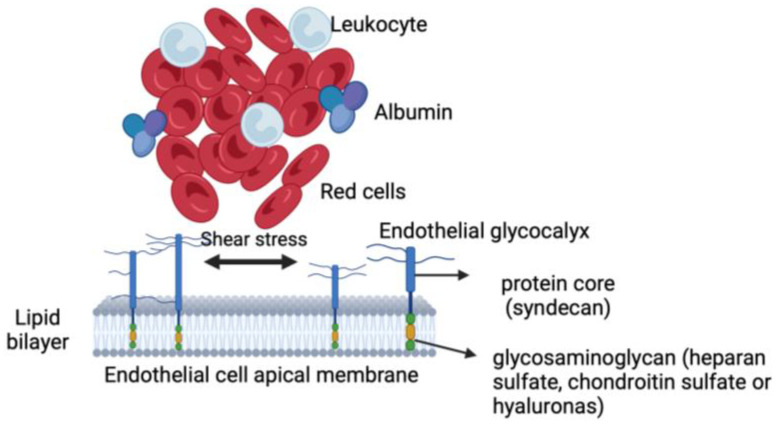
The endothelial glycocalyx is a proteoglycan complex, composed of a protein core (syndecans) and glycosaminoglycans, such as heparan sulfate, chondroitin sulfate, and hyaluronas, lining the luminal surface of endothelial cells (ECs) [22].

**Figure 5 ijms-26-01211-f005:**
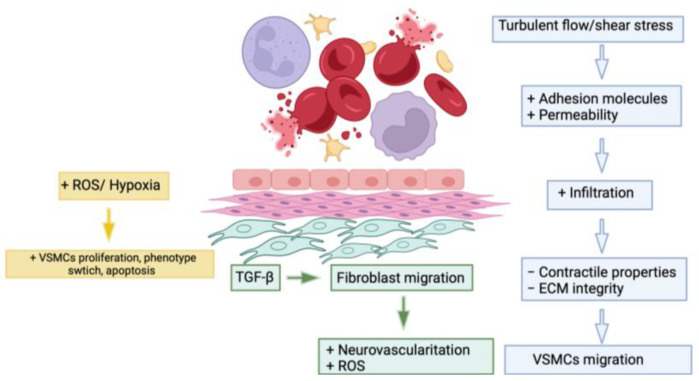
Alterations in the endothelial barrier responsible for the onset of aneurysmal pathology [27].

**Figure 6 ijms-26-01211-f006:**
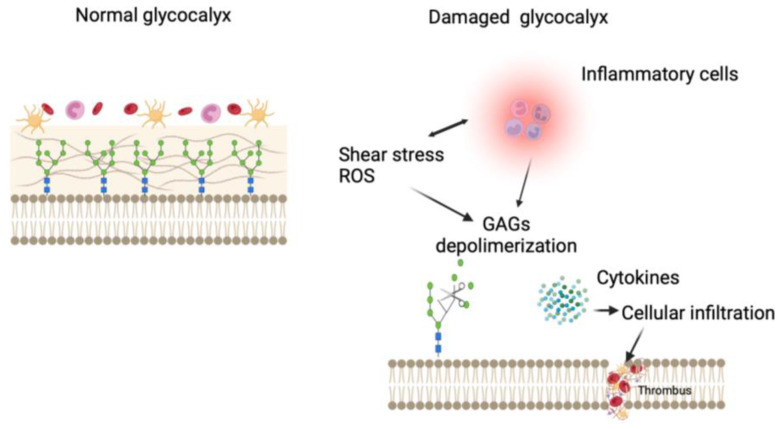
Differences between normal and damaged endothelial glycocalyx [21].

**Figure 7 ijms-26-01211-f007:**
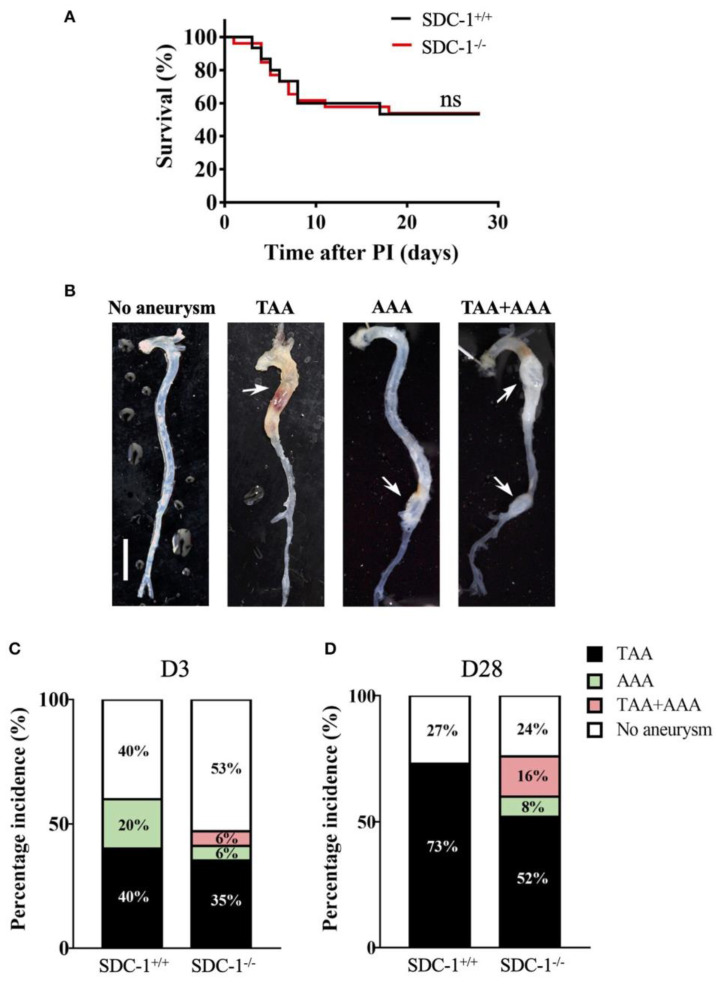
(**A**) Survival rate of mice after 28 days following Ang II infusion, compared with the Gehan–Breslow–Wilcoxon test. PI, pump implantation; SDC-1+/+: n = 15, SDC-1−/−: n = 26; ns, non-significant. (**B**) Aorta macroscopic images with/without aneurysms 28 days after Ang II infusion. Scale bar corresponds to 5 mm. (**C**,**D**) Percentage of TAA or AAA incidence in SDC-1+/+ or SDC-1−/− mice for 3 (**C**) or 28 days (**D**) of Ang II infusion. (**C**) SDC-1+/+: n = 5, SDC1−/−: n = 17. (**D**) SDC-1+/+: n = 15, SDC-1−/−: n = 26.

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
