# Peer review of "A Narrative Review: Syndecans in Aortic Aneurysm Pathogenesis and Course—Biomarkers and Targets?"

_ijms, 2025, doi:10.3390/ijms26031211_

Round 1

Reviewer 1 Report

Comments and Suggestions for Authors

This review analyses the role of glycocalyx in the genesis and progression of aortic aneurysms. The topic is covered with explanatory figures and tables which guide the reader in understanding the text.  With the growing awareness of the role of inflammation in the atherosclerotic process and the importance of recognizing its preclinical phase, the topic is interesting even if still confined to the preclinical phase. If I have to make a criticism I would advise the authors to develop a little more the generation of work hypotheses on the potential clinical use of glycolalyx markers. For example, they could improve the screening process for aortic aneurysms by integrating them with imaging techniques and/or other inflammatory/metabolic markers (e.g., high-sensitivity c-reactive protein). 

Suggested Ref:

Xu, C., Wang, G., Jin, G., Fei, X., Liu, C., Tang, L., ... & Yu, J. (2025). Genetic association between inflammatory factors and abdominal aortic aneurysm: Insights from a genome-wide association study. International Journal of Cardiology421, 132905.

Håland, A. B., Mattsson, E., Videm, V., Albrektsen, G., & Nyrønning, L. Å. (2025). Elevated High Sensitivity C-Reactive Protein and Risk of Abdominal Aortic Aneurysm: A Prospective Population Based Study in The Norwegian HUNT Study. European Journal of Vascular and Endovascular Surgery.

Author Response

Thank you for your review and suggestions. In this regard, we have expanded the section on the clinical use of syndecans, in particular on their role also in the clinical management of patients subject to trauma, and we have also further clarified our future proposal: to further encourage research on SDCs in order to further validate their value in the pathogenesis of aortic aneurysm, before moving on to a clinical application of the markers. Finally, we have included the references you proposed to further underline the usefulness of certain inflammatory biomarkers in the evaluation of the risk of abdominal aortic aneurysm. You will find the inserted parts written in red.

Reviewer 2 Report

Comments and Suggestions for Authors
  • Pisano et al. presented a narrative review focusing on syndecans, a family of proteoglycans, and their critical role in the pathogenesis of aortic aneurysms (AA). As essential components of the endothelial glycocalyx, syndecans contribute to the maintenance of aortic homeostasis. Their dysfunction, particularly through shedding and altered expression, has been implicated in endothelial damage by activating pro-inflammatory and growth-stimulating genes. These disruptions compromise the structural integrity of the glycocalyx, fostering pro-thrombotic states and contributing to both chronic and acute aortic damage associated with aneurysm development. The review emphasizes the significance of syndecans in the pathological mechanisms affecting the aortic wall, highlighting their potential as biomarkers and therapeutic targets.

    Overall, the manuscript is well-structured and offers a thorough analysis of syndecans in the context of aortic aneurysms. Below are a few comments and suggestions for improvement:

    1. Page 4, Line 150: Please expand on how pro-inflammatory cytokines facilitate the degradation of the endothelial glycocalyx. Providing detailed pathways or specific examples of cytokines involved, along with their direct or indirect effects on syndecan shedding and glycocalyx stability, would enhance the understanding of this mechanism.

    2. Page 10, Lines 351–354: Are there clinical studies that have evaluated the proposed biomarkers (syndecans or related molecules) in the blood of AA patients? Including such findings would strengthen the review by bridging the gap between basic research and clinical relevance.

Author Response

Thank you for your review.

Regarding your first comment, we have expanded the section on page 4, line 150 with the explanation between the correlation of the alteration of the endothelial glycocalyx and the altered production of NO (with the related alteration of the pathway), responsible for the alteration of the endothelial permeability.

Regarding the second comment, we have expanded the section on page 10, lines 351–354, with the integration of the manuscripts related to the use of certain biomarkers in patients with aortic aneurysm.

You will find the added sections written in red.

Round 2

Reviewer 1 Report

Comments and Suggestions for Authors

I have not further comments